# Retrospective analysis of the BariClip procedure: Clinical outcomes and complication profile

Saleh Abualhaj[1,2]*, Anas Alyazouri[3], Mosleh M. Abualhaj[4], Lina Alshadfan[5], Shadi Hamouri[1], Obada Alaraishy[3], Eman Alkhawaja[6], Amro Mureb[2], Ali Aloun[7], Abdallah Arabyat[8]

1 General Surgery Department, Faculty of Medicine, Al-Balqa Applied University, Salt, Jordan, 2 General Surgery Department, King Hussein Cancer Center (KHCC), Amman, Jordan, 3 General surgery department, Istiklal hospital Amman, Amman, Jordan, 4 Department of Networks and Cybersecurity, Al-Ahliyya Amman University, Amman, Jordan, 5 Al Balqa Applied University, Faculty of medicine, Department of pediatrics, Salt, Jordan, 6 Department of Nutrition and Food Technology, Al-Balqa Applied University, Salt, Jordan, 7 General Surgery, Royal Medical Services, Amman, Jordan, 8 Faculty of Medicine, Al-Balqa Applied University, Al-salt, Jordan

* Saleh.abualhaj@bau.edu.jo

## Abstract

### Background

Bariclip is an emerging non- resective bariatric device designed to restrict gastric capacity while preserving anatomical integrity. Unlike traditional sleeve gastrectomy, Bariclip implantation does not involve gastric resection, potentially minimizing surgical risk and allowing reversibility. However, data on its early efficacy and safety remain limited.

### Objective

To evaluate short-term surgical outcomes, weight loss metrics, and comorbidity resolution among patients undergoing Bariclip implantation.

### Methods

This retrospective observational study included 82 patients who underwent Bariclip placement at a single tertiary care center. Data were extracted from electronic medical records, operative logs, and follow-up notes. Outcomes assessed included total weight loss (TWL%) and excess weight loss (EWL%) at 2 weeks, 1, 2, 3, and 6 months postoperatively. Additional variables included operative time, hospital stay, early postoperative complications (within 30 days), reintervention rates, and changes in obesity-related comorbidities.

### Results

The cohort had a mean age of 37.6 ± 9.9 years, with the majority being female (76.8%) and obese (mean BMI = 36.6 ± 4.7 kg/m²). Most patients (91.5%) underwent

**Data availability statement:** The data supporting this study contain potentially identifying and sensitive information about participants. In accordance with institutional and national ethical standards, and as approved by the Research Ethics Committee at Al-Balqa Applied University (Ref:26/3/2/847) the data cannot be publicly shared to protect participant confidentiality and privacy. Requests for access to de-identified data may be directed to the Research Ethics Committee at Al-Balqa Applied University, which evaluates such requests on a case-by-case basis. Contact Information: Research Ethics Committee Al-Balqa Applied University Salt, Jordan Email: dsr@bau.edu.jo Phone: +962 5 349 1111.

**Funding:** The author(s) received no specific funding for this work.

**Competing interests:** The authors have declared that no competing interests exist.

surgery for obesity management. Postoperatively, patients reported low pain scores (mean = 5.2), with no need for opioid analgesia and early mobilization in 62.2%. Complication rates were low (3.6%). Repeated measures ANOVA revealed a significant reduction in BMI over time ($p < 0.001$). At 6 months, mean %TWL was $20.04\% \pm 5.39\%$ and mean %EWL reached $74.32\% \pm 40.75\%$. The most rapid weight loss occurred during the first three months, followed by a slower but consistent decline thereafter.

## Conclusion

Bariclip surgery demonstrated favorable short-term safety and efficacy, with substantial weight loss and low complications rate observed within six months. These findings support Bariclip as a promising minimally invasive option for weight management in select patient populations.

## Introduction

The global obesity epidemic affects over two billion individuals and has demonstrated a continuous and significant increase in prevalence since 1980 [1]. In Jordan, the burden of obesity is similarly concerning. According to the most recent national data, the prevalence of obesity among adults is estimated to be 35.5% in women and 23.6% in men, with over 60% of the adult population being overweight or obese [2,3]. This growing public health issue contributes substantially to the increasing rates of obesity-related comorbidities, such as type 2 diabetes, cardiovascular diseases, and metabolic syndrome, placing immense pressure on the healthcare system [4,5].

For patients with morbid obesity, bariatric surgery offers a durable means of achieving sustainable weight loss and ameliorating associated comorbidities. Over the past decade, Laparoscopic Sleeve Gastrectomy (LSG) has become the most frequently performed bariatric procedure worldwide [6]. However, a significant drawback of LSG is the high incidence of postoperative gastroesophageal reflux disease (GERD), which can manifest as de novo disease or an exacerbation of pre-existing symptoms, often leading to dependence on proton pump inhibitors (PPIs) [7]. A serious consequence of this chronic reflux is the potential development of Barrett's esophagus (BE), reported in up to 18% of post-LSG patients [8].

Despite demonstrated efficacy, the overall utilization of bariatric surgery remains limited, largely due to patient apprehension regarding permanent anatomical alterations. While LSG was initially perceived as less invasive than gastric bypass, the requisite "cutting" and resection of the stomach have hindered its widespread acceptance. This patient hesitancy has catalyzed interest in anatomy-sparing alternatives, leading to the development of various endoscopic procedures for gastric volume reduction [9]. However, the long-term durability of weight loss achieved with these endoscopic techniques remains a significant concern [10].

To address these challenges, a "non-resectional" gastric sleeve concept was developed and published in 2017 [11]. This innovation aimed to mitigate patient

concerns regarding the gastric resection and irreversibility inherent to LSG. The result was the BariClip, a device composed of a silicone-covered titanium body. The procedure involves the laparoscopic application of this vertical clip parallel to the lesser curvature of the stomach, as previously described [12].

The BariClip functions as a non-adjustable, restrictive device that mimics the effects of a sleeve gastrectomy without requiring stapling or resection. By preserving the gastric anatomy and leaving the digestive pathway unaltered, the procedure avoids risks such as staple line leaks or malabsorptive side effects and is potentially reversible [11]. The principal complications associated with the BariClip are device slippage and erosion. While erosion is primarily related to device properties and occlusion pressure, slippage is largely attributed to surgical placement and fixation methods. Consequently, operative techniques have continued to evolve to minimize the incidence of these adverse events [11–13].

This study aimed to evaluate the short-term safety and effectiveness of the Bariclip device as a novel restrictive bariatric procedure by analyzing weight loss outcomes—specifically total weight loss percentage (TWL%) and excess weight loss percentage (EWL%)—alongside postoperative complication rates in a cohort of patients undergoing the procedure at a single tertiary care center.

## Method

### Study design and setting

This study is a retrospective observational cohort analysis based on prospectively collected data from a single high-volume bariatric surgery center. It aims to evaluate early postoperative complications and short-term outcomes (at 6 months) in patients who underwent laparoscopic vertical clip gastroplasty (LBCG) using the nonadjustable BariClip system. The surgeries were performed at Istiklal Hospital, a private tertiary care facility, between July 1, 2023, and December 1, 2024, to allow for a minimum of 6 months of postoperative follow-up. Data collection for the study was conducted between June 8, 2025, and July 10, 2025.

### Study population

The study included 82 consecutive adult patients (aged ≥18 years) who underwent primary LBCG using the BariClip device during the study period. Inclusion Criteria: Adults aged 18 years and older, Underwent primary laparoscopic BariClip gastroplasty at Istiklal Hospital, prospectively documented perioperative and follow-up data, Minimum of 6 months postoperative follow-up. The exclusion Criteria: Revision or revisional bariatric surgeries, Incomplete clinical or follow-up records, Follow-up duration of less than 6 months.

### Surgical technique

All patients underwent laparoscopic Bariclip surgery under general anesthesia (GA). The procedure was performed by experienced bariatric surgeons using a standardized technique to ensure consistency across the cohort. (Fig 1)

Preoperatively, all patients received prophylactic antibiotics to minimize the risk of surgical site infection. Intraoperative anticoagulation was administered routinely as per the institutional protocol to reduce the risk of thromboembolic events.

The procedure was completed laparoscopically in all cases, with an average operative time of one hour. Blood loss during surgery was consistently minimal, estimated at less than 50 mL for all patients. No intraoperative complications were observed in this series, reflecting both the feasibility and safety of the laparoscopic Bariclip approach in this patient population.

Postoperative complications were recorded and classified. Any patient with suspected adverse events underwent imaging or endoscopic evaluation as indicated. Management decisions were based on clinical presentation and surgeon discretion.

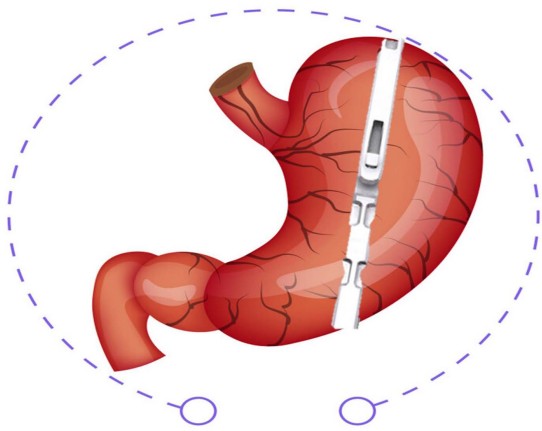

**Fig 1. Schematic illustration of laparoscopic vertical clip gastroplasty using the nonadjustable BariClip system, showing the placement of the clip along the stomach to create a restrictive effect while preserving the continuity of the gastrointestinal tract.** Figure generated by the authors.

## Surgeon and surgical standardization

All procedures were performed by a single experienced bariatric surgeon, ensuring consistency in the surgical technique and perioperative care across the entire patient cohort.

## Data collection

Data were prospectively collected as part of routine clinical care and retrospectively extracted from the hospital's electronic medical records, operative logs, and outpatient follow-up documentation. Collected data included demographic information such as age, sex, preoperative weight and body mass index (BMI), as well as obesity-related comorbidities including type 2 diabetes, hypertension, dyslipidemia, and obstructive sleep apnea. Preoperative baseline characteristics were recorded, including weight, BMI, and comorbidity status. Operative details encompassed the date of surgery, total operative time, intraoperative findings, any intraoperative complications, and length of hospital stay. In addition, early postoperative complications occurring within 30 days of surgery—such as bleeding, leak, infection, nausea, vomiting, and Bariclip slippage or migration—were documented. Hospital readmissions and reinterventions within the first 6 months were also tracked. The status of comorbidities at 6 months postoperatively was categorized as resolved, improved, or unchanged based on clinical evaluation and laboratory results.

## Outcome measurement

Postoperative symptoms including pain, nausea, and abdominal distention were assessed using a patient-reported numeric rating scale (NRS) ranging from 0 (no symptom) to 10 (worst possible symptom), as documented in the medical records within the first 24 hours after surgery. Additional early recovery indicators, such as analgesic consumption, time to first ambulation, and total hospital length of stay, were obtained from nursing and anesthesia records. Early complications were documented during the intraoperative period and throughout the hospital admission, with further follow-up conducted at 6 months to assess for any late complications, hospital readmissions, or the need for reinterventions. The primary outcome of the study was the percentage reduction in BMI at 6 months compared to baseline. Secondary outcomes included trends in weight and BMI at all postoperative timepoints, the rate and type of perioperative complications, early recovery outcomes (pain, nausea, vomiting, abdominal distention, analgesic use, time to ambulation, and length of stay), as well as the frequency and nature of readmissions or reinterventions occurring within the 6-month postoperative period.

## Calculated variables

To assess weight loss outcomes, two standard bariatric metrics were calculated: percent total weight loss (%TWL) and percent excess weight loss (%EWL). %TWL was defined as the percentage reduction from the initial weight, calculated using the formula:

$$\%TWL = [(\text{Initial Weight} - \text{Follow-up Weight})/\text{Initial Weight}] \times 100.$$

%EWL was determined using the formula:

$$\%EWL = [(\text{Initial Weight} - \text{Follow-up Weight})/(\text{Initial Weight} - \text{Ideal Body Weight})] \times 100,$$

where Ideal Body Weight was estimated based on a body mass index (BMI) of 25 (kg/m²) [14]. These metrics were computed at each follow-up timepoint (2 weeks, and 1, 2, 3, and 6 months postoperatively) to evaluate the progression and efficacy of weight loss following BariClip placement.

## Ethical approval

This study was conducted in accordance with the Declaration of Helsinki and approved by the Institutional Review Board (IRB) of Al-Balqa Applied University (BAU) [Ref:26/3/2/847]. Given the retrospective design and use of previously collected, de-identified data, the IRB granted a waiver of informed consent. All patient information was anonymized, and the dataset obtained from Istiklal Hospital was fully de-identified prior to analysis. No identifiable information was accessible to the research team at any point during or after data collection, in accordance with institutional privacy and ethical standards.

## Statistical analysis

Patient demographic and baseline clinical characteristics were summarized using descriptive statistics means and standard deviations (SD) for continuous variables, and frequencies and percentages for categorical variables.Prior to conducting parametric analyses, the normality of BMI data at each time point was assessed using the Shapiro-Wilk test and visual inspection of Q-Q plots. The results confirmed that BMI values were normally distributed across all time points, supporting the use of parametric tests.

Weight loss outcomes were primarily analyzed using an intention-to-treat (ITT) approach, including all patients who underwent Bariclip placement. Additionally, a sensitivity analysis was performed excluding the patient who required early clip removal due to gastric erosion, to assess the impact on overall weight loss results.

Changes in body mass index (BMI) over time were analyzed using a repeated measures analysis of variance (ANOVA). Mauchly's test of sphericity was performed to assess the sphericity assumption; when this assumption was violated ($p < 0.001$), the Greenhouse-Geisser correction was applied to adjust the degrees of freedom. Pairwise comparisons between time points were conducted with appropriate post hoc tests, and p-values were adjusted for multiple comparisons using the Bonferroni method. All analyses were conducted using *R software version 4.2.1*. A two-sided p-value of less than 0.05 was considered statistically significant.

## Patient and public involvement

This study did not involve patients or members of the public in the design, conduct, or dissemination of the research. No public representatives or patients were consulted for feedback on the research process, methodology, or the reporting of results.

## Results

### Study participants demographics and baseline characteristics

A total of 82 patients were included in the study, with a mean age of 37.6 years (SD 9.9; range 18–60). The majority were female (76.8%) and Jordanian nationals (95.1%). Approximately 59.8% were smokers, and 45.1% had at least one comorbidity, most commonly diabetes or insulin resistance (54.1% of those with comorbidities). Most patients underwent Bariclip surgery for obesity (91.5%), with the remainder for overweight management (8.5%). The mean baseline BMI was 36.6 kg/m² (SD 4.7; range 27.2–48.5). All procedures were performed laparoscopically, with a mean operative time of 60.4 minutes (SD 2.15; range 58–65 minutes). Estimated blood loss was minimal, averaging 35.8 mL (SD 10.5; range 10–50 mL), and no intraoperative complications were observed. Detailed baseline characteristics are summarized in Table 1.

### Postoperative outcomes

No patients required opioid analgesia during their hospital stay (0%). The mean postoperative pain score was **5.2** (SD 3.7; range 0–10). Mean scores for nausea and abdominal distention were **3.4** (SD 3.6; range 0–10) and **6.1** (SD 3.4; range 0–10), respectively. No patients experienced vomiting postoperatively (**0%**). Over half of the patients (**54.9%**) required mild analgesics (paracetamol) during admission, while the remainder (**45.1%**) did not require additional pain medication. The majority of patients ambulated within **1–2 hours** after surgery (**62.2%**), with a smaller proportion mobilizing after 3–4 hours. Hospital stay was either **1 day (52.4%)** or **2 days (47.6%)** for all patients.

Out of 82 patients, three (3.7%) developed early procedure-related complications before the 6-month follow-up, all of which required readmission and surgical intervention. Two patients (2.4%) experienced clip slippage—one at 3 weeks and the other at 2 months postoperatively—identified through follow-up imaging prompted by suboptimal weight loss and persistent vomiting. Both were successfully managed with laparoscopic clip repositioning. One patient (1.2%) developed gastric erosion at approximately 4 months postoperatively, diagnosed via endoscopy after presenting with epigastric pain and anemia. The clip was removed laparoscopically, and the patient recovered without further complications.

No postoperative complications were observed at 6-month follow-up. (Table 2)

### BMI trend over time

A repeated measures ANOVA (S1 Tabel in S1 File) with Greenhouse–Geisser (S3 Table in S1 File) correction demonstrated a statistically significant effect of time on BMI following Bariclip surgery ($F(1.51, 121.96) = 127.7$, $p < 0.001$, ges = 0.211). Mauchly's test (S2 Table in S1 File) indicated a violation of sphericity ($W = 0.000574$, $p < 0.001$); therefore, degrees of freedom were adjusted accordingly. Post hoc pairwise comparisons with Bonferroni correction revealed that mean BMI decreased significantly from baseline (Mean = 36.6, SD = 4.7) to all subsequent time points (2 weeks, 1 month, 2 months, 3 months, and 6 months; all $p < 0.001$) (S4 Table in S1 File). A clear downward trend was observed, with mean BMI reaching 29.2 (SD = 3.9) at 6 months postoperatively (see Fig 2 and Table 3).

### Weight loss outcomes

There was a progressive reduction in weight, %TWL, and %EWL across all timepoints. At 6 months postoperatively, the mean %TWL reached **20.04% ± 5.39%**, while the mean %EWL was **74.32% ± 40.75%**, indicating substantial weight reduction and improvement relative to excess body weight. The most significant drop occurred within the first three months, with continued but slower weight loss up to the six-month mark (Table 4 & Fig 3).

Exclusion of the patient who underwent clip removal did not significantly alter the overall trends in weight loss outcomes (see Supplementary S5 Table in S1 File), supporting the robustness of the primary findings.

**Table 1. Baseline Characteristics of the Study Cohort (N = 82).**

| Variable | n (%) or Mean (SD) |
|---|---|
| **Age** | |
| Mean (SD) | 37.6 (9.9) |
| Range | 18.0 - 60.0 |
| **Gender** | |
| F | 63 (76.8%) |
| M | 19 (23.2%) |
| **Nationality** | |
| Jordanian | 78 (95.1%) |
| Non-Jordanian | 4 (4.9%) |
| **Previous Surgical History** | |
| No | 74 (90.2%) |
| Yes | 8 (9.8%) |
| **Smoking Status** | |
| Non-Smoker | 33 (40.2%) |
| Smoker | 49 (59.8%) |
| **Marital Status** | |
| Married | 48 (58.5%) |
| Single | 34 (41.5%) |
| **Employment Status** | |
| Non-Employed | 20 (24.4%) |
| Employed | 62 (75.6%) |
| **Education level** | |
| University | 55 (67.1%) |
| Diploma | 16 (19.5%) |
| Secondary | 11 (13.4%) |
| **Daily PHYSICAL ACTIVITY level before surgery** | |
| Moderately active | 13 (15.9%) |
| Lightly active | 55 (67.1%) |
| Sedentary | 14 (17.1%) |
| **COMORBIDITIES** | |
| None | 45 (54.9%) |
| ≥1 Comorbidity | 37 (45.1%) |
| **Type of comorbidity (%within the 37 patients with ≥1 comorbidity)** | |
| Palpitation | 4 (10.8%) |
| Sleep apnea | 5 (13.5%) |
| HTN | 7 (18.9%) |
| DM or Insulin resistance | 20 (54.05%) |
| **Baseline Weight (Kg)** | |
| Mean (SD) | 101.9 (17.6) |
| Range | 71.0 - 146.0 |
| **Baseline Hight (m)** | |
| Mean (SD) | 1.7 (0.1) |
| Range | 1.5 - 1.9 |
| **Baseline BMI (kg/m²)** | |
| Mean (SD) | 36.6 (4.7) |
| Range | 27.2 - 48.5 |

*(Continued)*

**Table 1.** (Continued)

| Variable | n (%) or Mean (SD) |
|---|---|
| **Reason for Surgery** | |
| Obesity | 75 (91.5%) |
| Overweight | 7 (8.5%) |
| **Operative Approach** | |
| Laparoscopic | 82 (100%) |
| **Operative Time (minutes)** | |
| Mean (SD) | 60.4 (2.15) |
| Range | 58-65 |
| **Estimated Blood Loss (mL)** | |
| Mean (SD) | 35.8 (10.5) |
| Range | 10 - 50 |
| **Intraoperative Complications** | |
| No | 82 (100%) |

## Discussion

This study demonstrated favorable early outcomes for laparoscopic vertical clip gastroplasty (LBCG) using the BariClip. Over the 6-month follow-up period, patients experienced a significant reduction in body mass index (BMI), as well as marked improvements in percent total weight loss (%TWL) and percent excess weight loss (%EWL). The procedure was associated with a low complication rate, with only three patients (3.7%) experiencing early procedure-related complications: two cases (2.4%) of clip slippage and one case (1.2%) of gastric erosion. Importantly, postoperative recovery was smooth across the cohort, with no need for opioid analgesia and successful implementation of early mobilization protocols. These results suggest that Bariclip may be a safe and effective non-resectional alternative to traditional bariatric procedures in appropriately selected patients.

Our findings are consistent with and, in some cases, more favorable than previously reported data for other restrictive bariatric procedures. For instance, typical 6-month outcomes for laparoscopic adjustable gastric banding (LAGB) have demonstrated EWL rates of 25–49%, while laparoscopic sleeve gastrectomy (LSG) yields EWL rates around 33–63% at similar time points [15]. Amar et al. described single-incision and conventional laparoscopic adjustable gastric banding in adolescents, reporting median operative times of 106 minutes for single-incision laparoscopic surgery for adjustable gastric banding and 82 minutes for conventional laparoscopy for adjustable gastric banding, with hospitalization of 2–3 days and postoperative analgesia of 1.8–2.4 days, illustrating that even minimally invasive approaches require longer operative times and hospital stays compared with our BariClip cohort [16]. In comparison, Ko et al. reported LSG outcomes with a mean operative time of 109.6±35.9 minutes, hospital stay of 5.1±1.2 days, and early postoperative complications in 4% of patients [17]. In contrast, all BariClip procedures in our study were completed laparoscopically with a mean operative time of 60.4 minutes (SD 2.15; range 58–65 minutes), minimal estimated blood loss (35.8 mL), no intraoperative complications, and a hospital stay of 1–2 days.

These findings suggest that BariClip may offer advantages in operative efficiency, early recovery, and perioperative safety, although our dataset was limited to weight loss and perioperative outcomes. Systematic data on comorbidity resolution, GERD incidence, and quality-of-life measures were not available in this retrospective cohort, which represents a limitation of our study.

Nevertheless, insights from the broader bariatric literature provide useful context. LSG has consistently demonstrated substantial improvement in obesity-related comorbidities such as type 2 diabetes, hypertension, and dyslipidemia, though it carries an increased risk of de novo or worsened GERD [18,19]. LAGB generally achieves more modest comorbidity

**Table 2. Postoperative outcomes (N = 82).**

| | |
|---|---|
| **Opioid Use** | |
| No | 82 (100%) |
| **Postoperative Pain Score (0–10)** | |
| Mean (SD) | 5.2 (3.7) |
| Range | 0–10 |
| **Nausea Score (0–10)** | |
| Mean (SD) | 3.4 (3.6) |
| Range | 0–10 |
| **Abdominal Distention Score (0–10)** | |
| Mean (SD) | 6.1 (3.4) |
| Range | 0–10 |
| **Vomiting** | |
| No | 82 (100%) |
| **Use of mild Analgesics within the admission (Paracetamol)** | |
| No | 37 (45.1%) |
| Yes | 45 (54.9%) |
| **Time to Ambulation After Surgery (hours)** | |
| 1 | 24 (29.3%) |
| 2 | 27 (32.9%) |
| 3 | 18 (22.0%) |
| 4 | 13 (15.9%) |
| **Hospital Stay (days)** | |
| 1 | 43 (52.4%) |
| 2 | 39 (47.6%) |
| **Early Postoperative Complications (30-day)** | |
| Gastric Erosion | 1 (1.2%) |
| Clip Slippage | 2 (2.4%) |
| None | 79 (96.3%) |
| **6-Month Postoperative Complications** | |
| None | 82 (100%) |
| **Readmission/Reintervention** | |
| No | 79 (96.3%) |
| Yes | 3 (3.7%) |

resolution and is burdened by late device-related complications such as slippage, erosion, and reoperation [20]. Early Bar-iClip data from Bonaldi et al. similarly reported promising short-term weight loss but noted a non-negligible incidence of GERD symptoms during follow-up [21]. Future prospective studies with standardized collection of comorbidity and GERD outcomes will be essential to establish whether BariClip offers a more favorable risk–benefit profile compared with LAGB and LSG.

While both BariClip and LAGB are reversible, anatomy-preserving procedures, LAGB has been associated with device-related complications such as slippage, erosion, and explantation. A study by Gálvez-Valdovinos et al. reporting on 364 LAGB cases over five years observed esophagogastric obstruction (3.2%), intragastric erosion (54.5%), chronic band slippage (24.6%), gastric reservoir dilatation (6.2%), esophageal dilation (3%), device malfunction (5.1%), and slippage into the mediastinum (0.5%). These findings highlight that late complications are relatively common with LAGB [20]. Although long-term data for BariClip are not yet available, our short-term results demonstrate a favorable safety

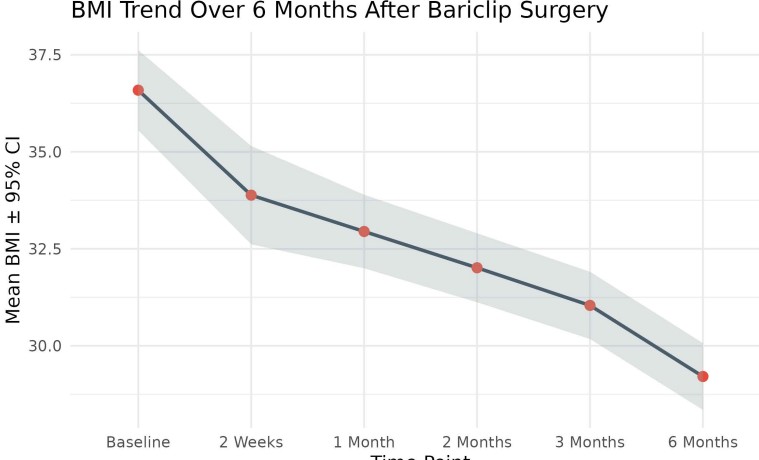

**Fig 2. Mean BMI trend across six time points after Bariclip surgery.**

**Table 3. Mean (SD) BMI and 95% confidence intervals for each postoperative follow-up point.**

| Timepoint | N | BMI Mean | SD | SE | 95% CI |
|---|---|---|---|---|---|
| Baseline | 82 | 36.58839 | 4.701144 | 0.5191546 | (35.56-37.62) |
| 2 Weeks | 82 | 33.88318 | 5.766706 | 0.6368262 | (32.62-35.15) |
| 1 Month | 82 | 32.94683 | 4.310477 | 0.4760125 | (32-33.89) |
| 2 Months | 82 | 32.01156 | 4.047517 | 0.4469734 | (31.12-32.9) |
| 3 Months | 82 | 31.0418 | 3.941986 | 0.4353195 | (30.18-31.91) |
| 6 Months | 82 | 29.21007 | 3.914937 | 0.4323324 | (28.35-30.07) |

**Table 4. Mean (SD) of Weight, TWL%, and EWL% for each postoperative follow-up point.**

| Time Point | Mean Weight (kg) | Mean %TWL±SD | Mean %EWL±SD |
|---|---|---|---|
| Baseline | 101.9 | – | – |
| 2 weeks | 95.6 | 6.12±1.83 | 22.87±13.04 |
| 1 month | 91.7 | 9.95±2.18 | 37.42±20.18 |
| 2 months | 89.1 | 12.43±3.18 | 46.05±24.43 |
| 3 months | 86.4 | 15.06±3.85 | 55.6±29 |
| 6 months | 81.3 | 20.04±5.39 | 74.32±40.75 |

profile and effective weight reduction. Future comparisons should focus on weight maintenance, resolution of obesity-related comorbidities, and incidence of gastroesophageal reflux disease (GERD), as these clinically important outcomes may distinguish BariClip from both LAGB and LSG, underscoring the need for further studies directly comparing these procedures.

In contrast, the Bariclip procedure in our cohort resulted in an average EWL of 74% at 6 months, and TWL 20%, highlighting its potential as a highly effective weight loss intervention in the short term. This is comparable to findings reported by Bonaldi et al., who analyzed outcomes in 149 patients and reported an average %TWL of 22.6% and a BMI reduction from 40±4.37 kg/m² to 28±4.29 kg/m² over the same period. Our complication rate was notably lower (3.7%) than theirs (8%), compared to 12 complications in Bonaldi's cohort, including higher reported rates of gastroesophageal reflux

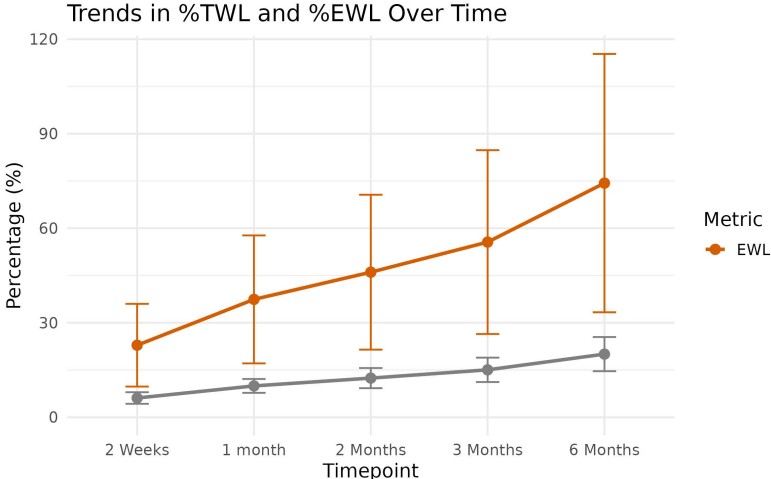

**Fig 3. Mean %TWL and %EWL across six time points after Bariclip surgery.**

disease (GERD). Additionally, none of our patients required opioids postoperatively, and all were mobilized early, highlighting the advantages of enhanced recovery pathways in our center. While Bonaldi et al. concluded that LBCG remains experimental, our findings further support its feasibility as a safe and reversible bariatric option in selected patients when performed in experienced centers with standardized protocols [21].

In addition to favorable weight loss outcomes, Bariclip was associated with minimal morbidity. Only 2 patients (2.4%) experienced clip slippage, and one patient (1.2%) developed gastric erosion—both recognized complications in restrictive bariatric procedures [12,22,23]. Notably, no cases of staple line leak or severe postoperative bleeding were reported, which are complications commonly associated with LSG [18,24,25]. Furthermore, no patient required opioid analgesia during hospitalization, and the incidence of postoperative nausea and vomiting was low. These findings reflect enhanced postoperative recovery, likely due to the non-resectional nature of the Bariclip and preservation of gastric anatomy [21].

The weight loss trajectory observed in our cohort followed a typical pattern seen in bariatric surgery, with the most rapid weight loss occurring during the first three to six months postoperatively, followed by a slower rate of reduction after that. This trend is in line with metabolic adaptation and lifestyle adjustments during the postoperative period [26]. Clinicians should be aware of this pattern when counseling patients and setting weight loss goals.

Bariclip may be particularly suitable for patients seeking a reversible and anatomy-preserving weight loss procedure. Its technical simplicity and short operative time offer additional advantages. Moreover, the low complication profile and avoidance of gastrointestinal stapling or rerouting may make it an attractive option for patients with relative contraindications to more invasive procedures. In this context, BariClip can be positioned in several ways: as an alternative to laparoscopic adjustable gastric banding (LAGB), given its reversibility without the long-term device-related complications commonly reported with bands; as a potential competitor to endoscopic sleeve gastroplasty (ESG), offering a surgical yet anatomy-preserving restrictive option that may provide greater durability than endoscopic approaches; and as a bridge procedure for high-risk or super-obese patients, where initial weight loss can reduce operative risk and enable later conversion to a more definitive resectional bariatric surgery if required [27]. Future comparative studies are needed to clarify which of these roles will be most clinically relevant.

Despite these promising results, our study has limitations. The retrospective design introduces the possibility of selection and information biases. The absence of a control group precludes direct comparison with standard procedures such as LSG or LAGB. Follow-up was limited to 6 months, which may not capture weight regain, late complications, or

long-term outcomes such as resolution of obesity-related comorbidities, GERD incidence, and quality-of-life measures. Additionally, detailed documentation of comorbidity improvement (e.g., type 2 diabetes, hypertension, dyslipidemia, and obstructive sleep apnea) was not routinely available in the patient records; therefore, our analysis was restricted to weight loss and perioperative parameters. We acknowledge this as a limitation and emphasize that future prospective studies are needed to evaluate these clinically important outcomes. Long-term prospective data will be critical to evaluating the durability, safety, and broader clinical impact of the BariClip procedure.

Nevertheless, our study has several strengths. It represents one of the largest single-center experiences with the BariClip procedure to date, providing valuable real-world data on early outcomes and safety in a high-volume bariatric surgery center. The inclusion of a relatively homogeneous patient cohort and standardized perioperative protocols enhances internal validity and allows for consistent evaluation of early postoperative metrics. Additionally, the comprehensive documentation of weight loss outcomes (%TWL, %EWL, BMI reduction), complication rates, and opioid-sparing postoperative recovery offers important insights into both the clinical efficacy and patient-centered aspects of recovery following LBCG. Finally, by integrating our findings with those of existing literature, this study contributes meaningfully to the growing body of evidence supporting the feasibility of BariClip as a promising, reversible alternative to traditional bariatric procedures.

## Conclusion

This study demonstrates that laparoscopic vertical clip gastroplasty (LBCG) using the nonadjustable BariClip is a safe and effective bariatric procedure in the short term, with significant reductions in BMI, %TWL, and %EWL observed over a 6-month period. The low incidence of postoperative complications, absence of opioid use, and rapid recovery highlight the potential advantages of this minimally invasive and reversible technique. Bariclip appears particularly suitable for patients with morbid obesity (BMI ≥ 35 kg/m²) or those with BMI ≥ 30 kg/m² with obesity-related comorbidities, who are motivated for lifestyle modification and prefer a reversible procedure; patients with severe gastroesophageal reflux, large hiatal hernia, or prior major gastric surgery may be less suitable.

With its favorable risk profile and potential for reversibility, Bariclip may represent an appealing option for patients who are not ideal candidates for permanent anatomical alteration. It may serve as a reversible, anatomy-preserving alternative to Lap-BAND, a minimally invasive competitor to ESG, or a bridge procedure for selected patients who are not candidates for permanent anatomical alterations.

Future prospective and randomized studies comparing Bariclip to LSG, endoscopic sleeve gastroplasty, or pharmacologic interventions are needed to confirm its place in the bariatric surgery spectrum. Moreover, assessment of comorbidity resolution, quality of life, and patient satisfaction will provide a more comprehensive evaluation of its clinical utility.

## Supporting information

**S1 File.    Table. ANOVA Table (Type III tests). S2 Table. Mauchly's Test for Sphericity. S3 Table. Sphericity Corrections. S4 Table. Post Hoc Pairwise Comparisons with Bonferroni. S5 Table. Sensitivity analysis of weight loss outcomes after exclusion of the patient who underwent clip removal.**
(ZIP)

## Author contributions

**Conceptualization:** Lina Alshadfan.

**Data curation:** Mosleh M. Abualhaj, Lina Alshadfan, Obada Alaraishy, Eman Alkhawaja.

**Formal analysis:** Saleh Abualhaj, Mosleh M. Abualhaj, Shadi Hamouri, Amro Mureb, Ali Aloun, Abdallah Arabyat.

**Investigation:** Anas Alyazouri, Abdallah Arabyat.

**Methodology:** Saleh Abualhaj.

**Project administration:** Anas Alyazouri, Mosleh M. Abualhaj, Shadi Hamouri, Eman Alkhawaja.

**Resources:** Obada Alaraishy.

**Software:** Ali Aloun.

**Supervision:** Ali Aloun.

**Validation:** Anas Alyazouri, Obada Alaraishy, Eman Alkhawaja.

**Visualization:** Amro Mureb.

**Writing – original draft:** Saleh Abualhaj, Lina Alshadfan.

**Writing – review & editing:** Anas Alyazouri, Mosleh M. Abualhaj, Shadi Hamouri, Eman Alkhawaja, Amro Mureb, Ali Aloun, Abdallah Arabyat.

## Acknowledgments

The authors thank Dr. Anas Alyazouri for performing all BariClip procedures included in this retrospective analysis and for his valuable clinical insights. We are also grateful to the medical-records team at Istiklal Hospital for their assistance in data retrieval. Ethical approval was granted by the Institutional Review Board of Al-Balqa Applied University, whose oversight is gratefully acknowledged.

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
