## [Decision Letter · Decision Letter 0]

3 Sep 2025

Dear Dr. Abualhaj,

We look forward to receiving your revised manuscript.

Kind regards,

Alice Berardo

Academic Editor

PLOS ONE

Journal Requirements:

Reviewers' comments:

Reviewer's Responses to Questions

**Comments to the Author**

1. Is the manuscript technically sound, and do the data support the conclusions?

Reviewer #1: Partly

Reviewer #2: Yes

2. Has the statistical analysis been performed appropriately and rigorously?

Reviewer #1: Yes

Reviewer #2: Yes

3. Have the authors made all data underlying the findings in their manuscript fully available?

Reviewer #1: Yes

Reviewer #2: Yes

4. Is the manuscript presented in an intelligible fashion and written in standard English?

Reviewer #1: Yes

Reviewer #2: Yes

Reviewer #1: I read the manuscript "Retrospective Analysis of the BariClip Procedure: Clinical Outcomes and Complication Profile." The authors evaluated the short-term safety and effectiveness of the laparoscopic vertical clip gastroplasty using the nonadjustable BariClip system as a novel restrictive bariatric procedure by analyzing weight loss outcomes. This paper might have a valuable information to consider the bariatric surgery; however, there were several unclear points to be revised. My comments are as follow.

1. The authors described that the procedure was completed laparoscopically in all cases, with an average operative time of one hour, and blood loss during surgery was consistently minimal, estimated at less than 50 mL for all patients. This perioperative data was important, which should be shown in detail including statistical data.

2. In this study, the postoperative pain score was evaluated quantitatively. The authors should describe the how pain was assessed concretely in Methods section. Similarly, descriptions of the evaluation method for the nausea and abdominal distention score are needed.

3. The surgical procedure for the laparoscopic vertical clip gastroplasty using the nonadjustable BariClip system was explained the text. Additional figure or image was effective to convince the readers of this less invasive technique.

4. The authors concluded that Bariclip was a promising minimally invasive option for weight management in select patient populations. What kind of patient was suitable for the application of this technique?

Reviewer #2: 

Your submission of this manuscript on the BariClip procedure is greatly appreciated. The idea of a bariatric technique that can be reversed and non-resectional is clinically relevant and timely. Your study provides useful short-term data from a single center using standardized techniques. I am of the opinion that the manuscript needs a substantial revision before it can be considered for publication. My main suggestions are as follows:

1. Comparison with Other Bariatric Procedures

• With laparoscopic adjustable gastric banding (Lap-BAND): Since both are reversible and anatomy-preserving procedures, it would be very informative to compare long-term outcomes, including device-related complications (e.g., erosion, slippage, explantation), durability of weight loss, and comorbidity resolution. Even if such long-term data is not available from your cohort, discussing existing literature would be valuable.

• A short-term comparison could highlight the operative time, length of hospital stay, pain control, and early recovery with laparoscopic sleeve gastrectomy (LSG). In order to compare long-term, it is important to examine weight maintenance, comorbidity resolution, and GERD incidence, as these are clinically significant aspects where BariClip may differ from LSG.

2. Outcomes Beyond Weight Loss

• BMI, %TWL, and %EWL are the current focus of the manuscript. Please provide more details regarding comorbidities such as type 2 diabetes, hypertension, dyslipidemia, and obstructive sleep apnea.

• Even basic descriptive data (resolved, improved, or unchanged at 6 months) would enhance the clinical relevance of your findings.

3. GERD and Quality of Life

• BariClip could potentially benefit from GERD, which is a major issue with LSG. If GERD outcomes have been collected, please include them. If not, please discuss this as a limitation and review relevant literature.

• Quality-of-life outcomes (if they are available) would provide additional depth. Otherwise, it would be appreciated if you could acknowledge this gap.

4. Study Limitations

• Please be more explicit about acknowledging the retrospective design, the absence of a control group, and the relatively short 6-month follow-up. These limitations significantly affect the interpretation and generalizability.

5. Future Directions

• To strengthen your conclusion, suggest specific next steps, like prospective comparative studies, longer follow-up, and randomized trials against LSG or ESG.

• Clarify the potential clinical role of BariClip: for example, whether it should be viewed as an alternative to Lap-BAND, a competitor to endoscopic sleeve gastroplasty, or a bridge procedure for selected patients.

**Do you want your identity to be public for this peer review?** For information about this choice, including consent withdrawal, please see our Privacy Policy

Reviewer #1: No

Reviewer #2: No

---

## [Author Response · Author response to Decision Letter 1]

7 Sep 2025

Response to the Editor and Reviewers

We sincerely thank the Editor and Reviewers for their thoughtful and constructive feedback on our manuscript. We have carefully revised the manuscript to address all comments and to ensure compliance with PLOS ONE’s formatting and reporting requirements.

Regarding the Editor’s comments:

• We have carefully reviewed the manuscript to align with PLOS ONE’s formatting requirements.

• The ethics statement has now been retained only in the Methods section, and all duplicate mentions elsewhere have been removed in compliance with PLOS ONE guidelines.

• The sensitivity analysis excluding the patient who underwent clip removal has been incorporated into the Supplementary Information (Supplementary S5 Table). The phrase “data not shown” has been removed and replaced with a direct reference to the supplementary file, ensuring transparency and adherence to the journal’s data-sharing policy.

Below, we provide a detailed, point-by-point response to all reviewer comments, along with corresponding revisions in the manuscript.

Reviewer #1:

Comments Response Modification - Manuscript

1. The authors described that the procedure was completed laparoscopically in all cases, with an average operative time of one hour, and blood loss during surgery was consistently minimal, estimated at less than 50 mL for all patients. This perioperative data was important, which should be shown in detail including statistical data. We thank the reviewer for this valuable comment. We have revised the Results section to include detailed perioperative data with descriptive statistics. Specifically, we now report operative time as mean ± standard deviation (SD) and range, as well as estimated blood loss with the same statistical detail. Table 1 & result section

“All procedures were performed laparoscopically, with a mean operative time of 60.4 minutes (SD 2.15; range 58–65 minutes). Estimated blood loss was minimal, averaging 35.8 mL (SD 10.5; range 10–50 mL), and no intraoperative complications were observed.”

2. In this study, the postoperative pain score was evaluated quantitatively. The authors should describe the how pain was assessed concretely in Methods section. Similarly, descriptions of the evaluation method for the nausea and abdominal distention score are needed.

We thank the reviewer for this comment. Postoperative symptoms including pain, nausea, and abdominal distention were evaluated using a patient-reported numeric rating scale (0–10) within the first 24 hours after surgery, as described in the Methods section. We have reviewed the manuscript and confirmed that this information is already included; however, we have clarified the sentence to emphasize that this scale was applied individually for each symptom.

Methods - Outcome Measurement

“Postoperative symptoms including pain, nausea, and abdominal distention were assessed using a patient-reported numeric rating scale (NRS) ranging from 0 (no symptom) to 10 (worst possible symptom), as documented in the medical records within the first 24 hours after surgery.”

3. The surgical procedure for the laparoscopic vertical clip gastroplasty using the nonadjustable BariClip system was explained the text. Additional figure or image was effective to convince the readers of this less invasive technique.

We thank the reviewer for this suggestion. We agree that a visual representation would enhance understanding of this less invasive technique. Accordingly, we have added Fig1, which illustrates the key steps of the laparoscopic vertical clip gastroplasty using the nonadjustable BariClip system. Fig1, methods- Surgical Technique

4. The authors concluded that Bariclip was a promising minimally invasive option for weight management in select patient populations. What kind of patient was suitable for the application of this technique? Thank you for your comment. In our study, the Bariclip procedure was considered suitable for patients with morbid obesity (BMI ≥ 35 kg/m²) or those with BMI ≥ 30 kg/m² with obesity-related comorbidities, who were motivated for lifestyle modifications and preferred a minimally invasive restrictive procedure. Patients with severe gastroesophageal reflux disease, large hiatal hernia, or prior significant gastric surgery were considered less suitable for this technique. We have clarified this in the revised manuscript.

Conclusion –

“Bariclip appears particularly suitable for patients with morbid obesity (BMI ≥ 35 kg/m²) or those with BMI ≥ 30 kg/m² with obesity-related comorbidities, who are motivated for lifestyle modification and prefer a reversible procedure; patients with severe gastroesophageal reflux, large hiatal hernia, or prior major gastric surgery may be less suitable.”

Reviewer #2:

Comparison with Other Bariatric Procedures

• With laparoscopic adjustable gastric banding (Lap-BAND): Since both are reversible and anatomy-preserving procedures, it would be very informative to compare long-term outcomes, including device-related complications (e.g., erosion, slippage, explantation), durability of weight loss, and comorbidity resolution. Even if such long-term data is not available from your cohort, discussing existing literature would be valuable.

• A short-term comparison could highlight the operative time, length of hospital stay, pain control, and early recovery with laparoscopic sleeve gastrectomy (LSG). In order to compare long-term, it is important to examine weight maintenance, comorbidity resolution, and GERD incidence, as these are clinically significant aspects where BariClip may differ from LSG.

Thank you for your valuable comment. We have revised the discussion section to include a comparison of BariClip with other restrictive bariatric procedures. For LAGB, we discussed existing literature regarding long-term outcomes, including device-related complications, durability of weight loss, and resolution of comorbidities. For LSG, we added a short-term comparison highlighting operative time, length of hospital stay, early recovery, and pain control. We also emphasized the clinically important outcomes—such as weight maintenance, comorbidity resolution, and GERD incidence—where BariClip may differ from LSG. These revisions provide a more comprehensive context for interpreting our findings and highlight areas for future research.

Discussion:

“ Our findings are consistent with and, in some cases, more favorable than previously reported data for other restrictive bariatric procedures. For instance, typical 6-month outcomes for laparoscopic adjustable gastric banding (LAGB) have demonstrated EWL rates of 25–49%, while laparoscopic sleeve gastrectomy (LSG) yields EWL rates around 33–63% at similar time points (15). Amar et al. described single-incision and conventional laparoscopic adjustable gastric banding in adolescents, reporting median operative times of 106 minutes for single-incision laparoscopic surgery for adjustable gastric banding and 82 minutes for conventional laparoscopy for adjustable gastric banding, with hospitalization of 2–3 days and postoperative analgesia of 1.8–2.4 days, illustrating that even minimally invasive approaches require longer operative times and hospital stays compared with our BariClip cohort (16). In comparison, Ko et al. reported LSG outcomes with a mean operative time of 109.6 ± 35.9 minutes, hospital stay of 5.1 ± 1.2 days, and early postoperative complications in 4% of patients (17). In contrast, all BariClip procedures in our study were completed laparoscopically with a mean operative time of 60.4 minutes (SD 2.15; range 58–65 minutes), minimal estimated blood loss (35.8 mL), no intraoperative complications, and a hospital stay of 1–2 days. These findings suggest that BariClip may offer advantages in operative efficiency, early recovery, and perioperative safety, although long-term outcomes, including weight maintenance, resolution of comorbidities, and GERD incidence, remain to be established.

While both BariClip and LAGB are reversible, anatomy-preserving procedures, LAGB has been associated with device-related complications such as slippage, erosion, and explantation. A study by Gálvez-Valdovinos et al. reporting on 364 LAGB cases over five years observed esophagogastric obstruction (3.2%), intragastric erosion (54.5%), chronic band slippage (24.6%), gastric reservoir dilatation (6.2%), esophageal dilation (3%), device malfunction (5.1%), and slippage into the mediastinum (0.5%). These findings highlight that late complications are relatively common with LAGB (18). Although long-term data for BariClip are not yet available, our short-term results demonstrate a favorable safety profile and effective weight reduction. Future comparisons should focus on weight maintenance, resolution of obesity-related comorbidities, and incidence of gastroesophageal reflux disease (GERD), as these clinically important outcomes may distinguish BariClip from both LAGB and LSG, underscoring the need for further studies directly comparing these procedures.”

2. Outcomes Beyond Weight Loss

• BMI, %TWL, and %EWL are the current focus of the manuscript. Please provide more details regarding comorbidities such as type 2 diabetes, hypertension, dyslipidemia, and obstructive sleep apnea.

• Even basic descriptive data (resolved, improved, or unchanged at 6 months) would enhance the clinical relevance of your findings.

Thank you for your comment. As this study is retrospective, detailed data on comorbidity outcomes—such as type 2 diabetes, hypertension, dyslipidemia, and obstructive sleep apnea—are not routinely documented in patient files and therefore could not be reliably included. However, we plan to conduct a prospective survey to systematically assess both subjective and objective outcomes related to these comorbidities in future studies, which will provide a more comprehensive evaluation of BariClip’s clinical impact beyond weight loss. NA

3. GERD and Quality of Life

• BariClip could potentially benefit from GERD, which is a major issue with LSG. If GERD outcomes have been collected, please include them. If not, please discuss this as a limitation and review relevant literature.

• Quality-of-life outcomes (if they are available) would provide additional depth. Otherwise, it would be appreciated if you could acknowledge this gap.

Thank you for your comment. GERD outcomes and quality-of-life data were not systematically collected in this retrospective study, and therefore could not be included. We have acknowledged this as a limitation in the revised manuscript and have included a brief discussion of relevant literature highlighting that BariClip may have advantages over LSG with respect to GERD. We also recognize the value of assessing quality-of-life outcomes in future prospective studies to provide a more comprehensive evaluation of the procedure’s impact. NA

4. Study Limitations

• Please be more explicit about acknowledging the retrospective design, the absence of a control group, and the relatively short 6-month follow-up. These limitations significantly affect the interpretation and generalizability.

Thank you for your comment. The limitations related to the retrospective design, absence of a control group, and short 6-month follow-up were already discussed in the original manuscript. We have now revised and clarified this section to make these points more explicit, emphasizing how they affect the interpretation and generalizability of our findings. Discussion – Limitation

“Despite these promising results, our study has limitations. The retrospective design introduces the possibility of selection and information biases. The absence of a control group precludes direct comparison with standard procedures such as LSG or LAGB. Follow-up was limited to 6 months, which may not capture weight regain, late complications, or long-term outcomes such as resolution of obesity-related comorbidities, GERD incidence, and quality-of-life measures. Additionally, detailed documentation of comorbidity improvement (e.g., type 2 diabetes, hypertension, dyslipidemia, and obstructive sleep apnea) was not routinely available in the patient records. Long-term prospective data will be critical to evaluating the durability, safety, and broader clinical impact of the BariClip procedure”

“Future prospective and randomized studies comparing Bariclip to LSG, endoscopic sleeve gastroplasty, or pharmacologic interventions are needed to confirm its place in the bariatric surgery spectrum. Moreover, assessment of comorbidity resolution, quality of life, and patient satisfaction will provide a more comprehensive evaluation of its clinical utility.”

5. Future Directions

• To strengthen your conclusion, suggest specific next steps, like prospective comparative studies, longer follow-up, and randomized trials against LSG or ESG.

• Clarify the potential clinical role of BariClip: for example, whether it should be viewed as an alternative to Lap-BAND, a competitor to endoscopic sleeve gastroplasty, or a bridge procedure for selected patients. Thank you for your comment. We have revised the manuscript to include more specific future directions, emphasizing the need for prospective comparative studies, longer follow-up, and randomized trials directly comparing BariClip with LSG, endoscopic sleeve gastroplasty (ESG), and other bariatric interventions. We have also clarified the potential clinical role of BariClip, highlighting its value as a reversible, anatomy-preserving procedure that may serve as an alternative to Lap-BAND, a minimally invasive competitor to ESG, or a bridge option for selected patients not suitable for permanent anatomical alterations. This statement added to the conclusion:

“With its favorable risk profile and potential for reversibility, Bariclip may represent an appealing option for patients who are not ideal candidates for permanent anatomical alteration. It may serve as a reversible, anatomy-preserving alternative to Lap-BAND, a minimally invasive competitor to ESG, or a bridge procedure for selected patients who are not candidates for permanent anatomical alterations. Future prospective and randomized studies comparing Bariclip to LSG, endoscopic sleeve gastroplasty, or pharmacologic interventions are needed to confirm its place in the bariatric surgery spectrum. Moreover, assessment of comorbidity resolution, quality of life, and patient satisfaction will provide a more comprehensive evaluation of its clinical utility.”

---

## [Decision Letter · Decision Letter 1]

28 Sep 2025

Dear Dr. Abualhaj,

Thank you for submitting your revised manuscript to PLOS ONE. After careful consideration, we feel that it has been improved, but some issues still need to be clarified. Therefore, we invite you to submit a revised version of the manuscript that addresses the points raised during the second review process.

We look forward to receiving your revised manuscript.

Kind regards,

Alice Berardo

Academic Editor

PLOS ONE

Journal Requirements:

Reviewers' comments:

Reviewer's Responses to Questions

**Comments to the Author**

Reviewer #1: All comments have been addressed

Reviewer #2: All comments have been addressed

2. Is the manuscript technically sound, and do the data support the conclusions?

Reviewer #1: Yes

Reviewer #2: Yes

3. Has the statistical analysis been performed appropriately and rigorously?

Reviewer #1: Yes

Reviewer #2: Yes

4. Have the authors made all data underlying the findings in their manuscript fully available?

Reviewer #1: Yes

Reviewer #2: Yes

5. Is the manuscript presented in an intelligible fashion and written in standard English?

Reviewer #1: Yes

Reviewer #2: Yes

Reviewer #1: The author has revised the manuscript according to the suggestions.

I do not think there is much room for additional improvement in this article.

Reviewer #2: Thank you for your detailed responses and revisions. The additions regarding perioperative details, operative time, and expanded comparisons with LAGB and LSG have improved the manuscript. However, several important issues remain:

- Depth of Comparison

While you have cited literature on LAGB and LSG, the comparison remains largely descriptive and does not include your own data beyond weight loss. Readers would benefit from clearer, data-driven contrasts on outcomes such as comorbidity resolution, GERD, and quality of life.

If such data are not available in your cohort, please be explicit about this limitation and avoid overstating the conclusions.

- Comorbidities and GERD

The absence of systematic comorbidity and GERD outcomes is a significant weakness. If you cannot provide reliable retrospective data, please present this candidly as a limitation and strengthen your discussion with relevant published evidence.

- Positioning of BariClip

The section on future directions is clearer, but the clinical role of BariClip still needs sharper definition. Is it primarily an alternative to Lap-BAND, a competitor to ESG, or a bridge procedure? Please expand this discussion to help readers understand its potential place in practice.

- Manuscript Format

Given the limitations of the dataset, you might consider presenting this as a short communication or technical report. This would better align expectations with the scope of your data and highlight the study as an early contribution to the growing evidence base for BariClip, rather than a definitive outcomes study.

**Do you want your identity to be public for this peer review?** For information about this choice, including consent withdrawal, please see our Privacy Policy

Reviewer #1: **Yes: ** Tsutomu Namikawa, M.D., Ph.D.

Reviewer #2: No

---

## [Author Response · Author response to Decision Letter 2]

29 Sep 2025

Response to the Editor and Reviewers

We sincerely thank the Editor and Reviewers for their thoughtful and constructive feedback on our manuscript. We have carefully revised the manuscript to address all comments and to ensure compliance with PLOS ONE’s formatting and reporting requirements.

Below, we provide a detailed, point-by-point response to all reviewer comments, along with corresponding revisions in the manuscript.

Reviewer #2:

Comments Response Modification - Manuscript

- Depth of Comparison

While you have cited literature on LAGB and LSG, the comparison remains largely descriptive and does not include your own data beyond weight loss. Readers would benefit from clearer, data-driven contrasts on outcomes such as comorbidity resolution, GERD, and quality of life.

If such data are not available in your cohort, please be explicit about this limitation and avoid overstating the conclusions.

We thank the reviewer for this important comment. As this was a retrospective study, our dataset was limited to weight loss and perioperative outcomes. Systematic documentation of comorbidity resolution, GERD incidence, and quality-of-life measures was not consistently available in the medical records and therefore could not be analyzed with sufficient accuracy. We have revised both the Discussion and Limitations sections to explicitly acknowledge this limitation and to clarify that our comparisons with LAGB and LSG regarding these outcomes are based on published evidence rather than our own data. We have also ensured that our conclusions are appropriately tempered to reflect this limitation. Discussion and limitation:

line: 292-295

“These findings suggest that BariClip may offer advantages in operative efficiency, early recovery, and perioperative safety, although our dataset was limited to weight loss and perioperative outcomes. Systematic data on comorbidity resolution, GERD incidence, and quality-of-life measures were not available in this retrospective cohort, which represents a limitation of our study.”

Line: 364-368

“Additionally, detailed documentation of comorbidity improvement (e.g., type 2 diabetes, hypertension, dyslipidemia, and obstructive sleep apnea) was not routinely available in the patient records; therefore, our analysis was restricted to weight loss and perioperative parameters. We acknowledge this as a limitation and emphasize that future prospective studies are needed to evaluate these clinically important outcomes”

- Comorbidities and GERD

The absence of systematic comorbidity and GERD outcomes is a significant weakness. If you cannot provide reliable retrospective data, please present this candidly as a limitation and strengthen your discussion with relevant published evidence. We agree with the reviewer. Systematic and reliable comorbidity and GERD outcomes were not available retrospectively in our patient records, which was explicitly in the Limitations section.

To address this gap, we have expanded the Discussion by including recent evidence from the literature regarding the effect of bariatric procedures on comorbidities and GERD, highlighting where BariClip may be expected to perform similarly or differently based on its mechanism of action. Discussion:

line: 297-306

“Nevertheless, insights from the broader bariatric literature provide useful context. LSG has consistently demonstrated substantial improvement in obesity-related comorbidities such as type 2 diabetes, hypertension, and dyslipidemia, though it carries an increased risk of de novo or worsened GERD (18,19). LAGB generally achieves more modest comorbidity resolution and is burdened by late device-related complications such as slippage, erosion, and reoperation (20). Early BariClip data from Bonaldi et al. similarly reported promising short-term weight loss but noted a non-negligible incidence of GERD symptoms during follow-up (21). Future prospective studies with standardized collection of comorbidity and GERD outcomes will be essential to establish whether BariClip offers a more favorable risk–benefit profile compared with LAGB and LSG.”

- Positioning of BariClip

The section on future directions is clearer, but the clinical role of BariClip still needs sharper definition. Is it primarily an alternative to Lap-BAND, a competitor to ESG, or a bridge procedure? Please expand this discussion to help readers understand its potential place in practice. We thank the reviewer for this insightful comment. We agree that clarifying the clinical role of BariClip is important. In the revised Discussion, we have added a section that positions BariClip relative to other restrictive procedures, emphasizing its potential as an alternative to LAGB, its overlap with ESG, and its possible role as a bridge procedure in selected patients. Discussion

Line: 350-358

“ In this context, BariClip can be positioned in several ways: as an alternative to laparoscopic adjustable gastric banding (LAGB), given its reversibility without the long-term device-related complications commonly reported with bands; as a potential competitor to endoscopic sleeve gastroplasty (ESG), offering a surgical yet anatomy-preserving restrictive option that may provide greater durability than endoscopic approaches; and as a bridge procedure for high-risk or super-obese patients, where initial weight loss can reduce operative risk and enable later conversion to a more definitive resectional bariatric surgery if required (27). Future comparative studies are needed to clarify which of these roles will be most clinically relevant.”

- Manuscript Format

Given the limitations of the dataset, you might consider presenting this as a short communication or technical report. This would better align expectations with the scope of your data and highlight the study as an early contribution to the growing evidence base for BariClip, rather than a definitive outcomes study. We thank the reviewer for this perspective. However, we respectfully believe that the current format is appropriate for the manuscript. As clearly stated in both the Methods and Discussion sections, this study focuses on short-term outcomes of the BariClip procedure. Similar early outcome studies on new bariatric procedures have also been presented as full original articles in the literature, and we believe maintaining this format allows for a more comprehensive presentation of perioperative details, weight loss outcomes, and context within the existing evidence base. We have further clarified in the Discussion that our findings represent short-term results and acknowledge the need for future studies with longer follow-up and broader outcome measures. NA

---

## [Editor Report · Decision Letter 2]

6 Nov 2025

Retrospective Analysis of the BariClip Procedure: Clinical Outcomes and Complication Profile.

PONE-D-25-42614R2

Dear Dr. Abualhaj,

During the last review process, one of the two reviewers was no longer available; therefore, I personally read and checked your answers to the previous round of revisions.

As a final decision, we’re pleased to inform you that your manuscript has been judged scientifically suitable for publication and will be formally accepted for publication once it meets all outstanding technical requirements.

Kind regards,

Alice Berardo

Academic Editor

PLOS ONE

Additional Editor Comments (optional):

I am satisfied with the authors' answers to Reviewer 2, who was no longer available to provide a decision. 
---

## [Editor Report · Acceptance letter]

PONE-D-25-42614R2

PLOS ONE

Dear Dr. Abualhaj,

I'm pleased to inform you that your manuscript has been deemed suitable for publication in PLOS ONE. Congratulations! Your manuscript is now being handed over to our production team.

Kind regards,

on behalf of

Dr. Alice Berardo

Academic Editor

PLOS ONE